# CONTINUAL LEARNING VIA WINNING SUBNETWORKS THAT ARISE THROUGH STOCHASTIC LOCAL COMPETITION

## ABSTRACT

The aim of this work is to address catastrophic forgetting in class-incremental learning. To this end, we propose deep networks that comprise blocks of units that compete locally to win the representation of each arising new task; competition takes place in a stochastic manner. This type of network organization results in sparse task-specific representations from each network layer; the sparsity pattern is obtained during training and is different among tasks. Under this rationale, our continual task learning algorithm regulates gradient-driven weight updates for each unit in a block on the grounds of winning probability. During inference, the network retains only the winning unit and zeroes-out all weights pertaining to non-winning units for the task at hand. As we empirically show, our method produces state-of-the-art predictive accuracy on few-shot image classification experiments, and imposes a considerably lower computational overhead compared to the current state-of-the-art.

## 1 INTRODUCTION

Continual Learning (CL), also referred to as Lifelong Learning (Thrun, 1995), aims to learn sequential tasks and acquire new information while preserving knowledge from previous learned tasks (Thrun & Mitchell, 1995). This paper's focus is on a variant of CL dubbed class-incremental learning (CIL) (Belouadah & Popescu, 2019; Gupta et al., 2020; Deng et al., 2021). The main principle of CIL is a CL scenario where on each iteration we are dealing with data from a specific task, and each task contains new classes that must be learnt. The addressed problem poses the major challenge of preventing catastrophic forgetting (McCloskey & Cohen, 1989): as network parameters get modified to allow for learning the new classes, the training process tends to overwrite old knowledge about classes presented many iterations ago. This constitutes a key obstacle to achieving human-level intelligence.

Recently, researchers have proposed various approaches to solve catastrophic forgetting during CIL, which can be organized in three main categories: (1) *regularization-based* solutions that utilize knowledge distillation (Hinton et al., 2015; Li & Hoiem, 2017) or penalize changes in weights deemed important for previous tasks (Kirkpatrick et al., 2017; Zenke et al., 2017; Aljundi et al., 2018); (2) *replay-based* methods that store in memory a collection of samples from previous tasks and replay them to retain the previous learned knowledge (Rebuffi et al., 2017; Saha et al., 2021), and (3) *architecture-based* methods that split the dense model into task-related modules for knowledge transfer and acquisition (Rusu et al., 2016; Hung et al., 2019).

Although the aforementioned CIL methods suppress catastrophic forgetting over sequentially arriving tasks, they often come at the price of increasing model size and computational footprint as new tasks arrive. To overcome this challenge, different research groups have recently drawn inspiration from the lottery ticket hypothesis (LTH) (Frankle & Carbin, 2019) to introduce the lifelong tickets (LLT) method (Chen et al., 2021), the Winning SubNetworks (WSN) method (Kang et al., 2022), and, more recently, the Soft-SubNetworks approach (Kang et al., 2023).

However, these recent advances are confronted with major limitations: (i) LLT entails an iterative pruning procedure, that requires multiple repetitions of the training algorithm for each task; (ii) the existing alternatives do not take into consideration the uncertainty in the used datasets, which would

benefit from the subnetwork selection process being stochastic, as opposed to hard unit pruning. In fact, it has been recently shown that stochastic competition mechanisms among locally competing units can offer important generalization capacity benefits for deep networks used in as diverse challenges as adversarial robustness (Panousis et al., 2021), video-to-text translation (Voskou et al., 2021), and model-agnostic meta-learning (Kalais & Chatzis, 2022).

Inspired from these facts, this work proposes a radically different regard toward addressing catastrophic forgetting in CIL. Our approach is founded upon the framework of stochastic local competition which is implemented in a task-wise manner. Specifically, our proposed approach relies upon the following novel contributions:

- **Task-specific sparsity in the learned representations.** We propose a novel mechanism that inherently learns to extract sparse task-specific data representations. Specifically, each layer of the network is split into blocks of competing units; local competition is stochastic and it replaces traditional nonlinearities, e.g. ReLU. Being presented with a new task, each block learns a distribution over its units that governs which unit specializes in the presented task. We dub this type of nonlinear units as *task winner-takes-all (TWTA)*. Under this scheme, the network learns a Categorical posterior over the competing block units; this is the winning unit posterior of the block. Only the winning unit of a block generates a non-zero output fed to the next network layer. This renders sparse the generated representations, with the sparsity pattern being task-specific.

- **Weight training strength regulation driven from the learned stochastic competition posteriors.** The network learns a global weight matrix that is not specific to a task, but evolves over time. During training, the network utilizes the learned Categorical posteriors over winning block units to dampen the update signal for weights that pertain to units with low winning posterior for the task at hand. In a sense, the block units with high winning posterior tend to get a stronger weight training cycle.

- **Winner-based weight pruning at inference time.** During inference for a given task, we use the (Categorical) winner posteriors learned for the task to select the winner unit of each block; we zero-out the remainder block units. This forms a *task-winning ticket* used for inference. This way, the size of the network used at inference time is significantly reduced; pruning depends on the number of competing units per block, since we drop all block units except for the selected winner with maximum winning posterior.

We evaluate our approach, dubbed TWTA for CIL (*TWTA-CIL*), on image classification problems. We show that, compared to the current state-of-the-art methods in the field: (i) it offers a considerable improvement in generalization performance, and (ii) it produces this performance with a network architecture that imposes a significantly lower memory footprint and better computational efficiency.

The remainder of this paper is organized as follows: In Section 2, we introduce our approach and describe the related training and inference processes. Section 3 briefly reviews related work. In Section 4, we perform an extensive experimental evaluation and ablation study of the proposed approach. In the last Section, we summarize the contribution of this work.

## 2 PROPOSED APPROACH

### 2.1 PROBLEM DEFINITION

CIL objective is to learn a unified classifier from a sequential stream of data comprising different tasks that introduce new classes. CIL methods should scale to a large number of tasks without immense computational and memory growth. Let us consider a CIL problem $T$ which consists of a sequence of $n$ tasks, $T = \{(C^{(1)}, D^{(1)}), (C^{(2)}, D^{(2)}), \ldots, (C^{(n)}, D^{(n)})\}$. Each task $t$ contains data $D^{(t)} = (\boldsymbol{x}^{(t)}, \boldsymbol{y}^{(t)})$ and new classes $C^{(t)} = \{c_{m_{t-1}+1}, c_{m_{t-1}+2}, \ldots, c_{m_t}\}$, where $m_t$ is the number of presented classes up to task $t$. We denote as $\boldsymbol{x}^{(t)}$ the input features, and as $\boldsymbol{y}^{(t)}$ the one-hot label vector corresponding to $\boldsymbol{x}^{(t)}$.

When training for the $t$-th task, we mainly use the data of the task, $D^{(t)}$. For additional regularization purposes, this dataset may be augmented with few exemplars from the previous $t-1$ tasks, retained

in a memory buffer of limited size (Rebuffi et al., 2017; Castro et al., 2018). We consider learners-classifiers that are deep networks parameterized by weights $\boldsymbol{W}$, and we use $f(\boldsymbol{x}^{(t)}; \boldsymbol{W})$ to indicate the output Softmax logits for a given input $\boldsymbol{x}^{(t)}$. Facing a new dataset $D^{(t)}$, the model's goal is to learn new classes and maintain performance over old classes.

## 2.2 MODEL FORMULATION

Let us denote as $\boldsymbol{x}^{(t)} \in \mathbb{R}^E$ an input representation vector presented to a dense ReLU layer of a traditional deep neural network, with corresponding weights matrix $\boldsymbol{W} \in \mathbb{R}^{E \times K}$. The layer produces an output vector $\boldsymbol{y}^{(t)} \in \mathbb{R}^K$, which is fed to the subsequent layers.

In our approach, a group of $J$ ReLU units is replaced by a group of $J$ competing linear units, organized in one block; each layer contains $I$ blocks of $J$ units. Within each block, different units are specialized in different tasks; only one block unit specializes in a given task $t$. The layer input is now presented to each block through weights that are organized into a three-dimensional matrix $\boldsymbol{W} \in \mathbb{R}^{E \times I \times J}$. Then, the $j$-th ($j = 1, \ldots, J$) competing unit within the $i$-th ($i = 1, \ldots, I$) block computes the sum $\sum_{e=1}^{E}(w_{e,i,j}) \cdot x_e^{(t)}$.

Fig. 1 illustrates the operation of the proposed architecture when dealing with task $t$. As we observe, for each task only one unit (the "winner") in a TWTA block will present its output to the next layer during forward passes through the network; the rest are zeroed out. During backprop (training), the strength of the updating signal is regulated from the relaxed (continuous) outcome of the competition process; this is encoded into a (differentiable) sample from the postulated Gumbel-Softmax.

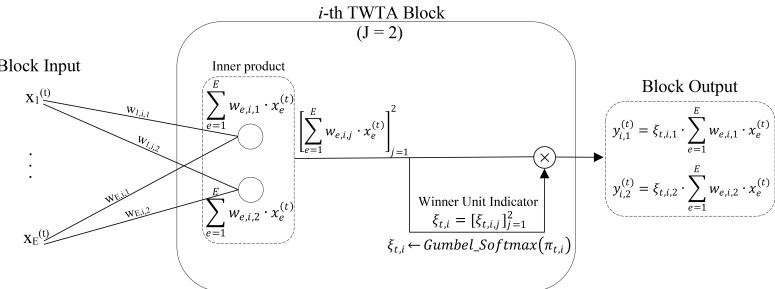

Figure 1: A detailed graphical illustration of the $i$-th block of a proposed TWTA layer (Section 2.2); for demonstration purposes, we choose $J = 2$ competing units per block. Inputs $\boldsymbol{x}^{(t)} = \{x_1^{(t)}, \ldots, x_E^{(t)}\}$ are presented to each unit in the $i$-th block, when training on task $t$. Due to the TWTA mechanism, during forward passes through the network, only one competing unit propagates its output to the next layer; the rest are zeroed-out.

Let us consider the $i$-th block in a TWTA layer, comprising $J$ units. We introduce the hidden winner indicator vector $\boldsymbol{\xi}_{t,i} = [\xi_{t,i,j}]_{j=1}^{J}$ of the $i$-th block pertaining to the $t$-th task. It holds $\xi_{t,i,j} = 1$ if the $j$-th unit in the $i$-th block has specialized in the $t$-th task (winning unit), $\xi_{t,i,j} = 0$ otherwise. We also denote $\boldsymbol{\xi}_t \in \{0, 1\}^{I \cdot J}$ the vector that holds all the $\boldsymbol{\xi}_{t,i} \in \{0, 1\}^J$ subvectors.

On this basis, the output of the layer, $\boldsymbol{y}^{(t)} \in \mathbb{R}^{I \cdot J}$, in our approach is composed of $I$ sparse subvectors $\boldsymbol{y}_i^{(t)} \in \mathbb{R}^J$. Succinctly, we can write $\boldsymbol{y}_i^{(t)} = [y_{i,j}^{(t)}]_{j=1}^{J}$, where:

$$y_{i,j}^{(t)} = \xi_{t,i,j} \sum_{e=1}^{E}(w_{e,i,j}) \cdot x_e^{(t)} \in \mathbb{R} \tag{1}$$

We postulate that the hidden winner indicator variables are drawn from a Categorical posterior distribution that yields:

$$p(\boldsymbol{\xi}_{t,i}) = \text{Categorical}(\boldsymbol{\xi}_{t,i} | \boldsymbol{\pi}_{t,i}) \tag{2}$$

The hyperparameters $\boldsymbol{\pi}_{t,i}$ are optimized during model training, as we explain next. The network learns the global weight matrix $\boldsymbol{W}$, that is not specific to a task, but evolves over time. During training, by learning different winning unit distributions, $p(\boldsymbol{\xi}_{t,i})$, for each task, we appropriately

mask large parts of the network, dampen the training signal strength for these parts, and mainly direct the training signal to update the fraction of $W$ that pertains to the remainder of the network. We argue that this asymmetric weight updating scheme during backpropagation, which focuses on a small winning subnetwork, yields a model less prone to catastrophic forgetting.

In Fig. 1, we depict the operation principles of our proposed network. As we show therein, the winning information is encoded into the trained posteriors $p(\boldsymbol{\xi}_{t,i})$, which are used to regulate weight training, as we explain in Section 2.4. This is radically different from Chen et al. (2021), as we do not search for optimal winning tickets during CIL via repetitive pruning and retraining for each arriving task. This is also radically different from Kang et al. (2023), where a random uniform mask is drawn for regulating which weights will be updated, and another mask is optimized to select the subnetwork specializing to the task at hand. Instead, we perform a single backward pass to update the winning unit distributions, $p(\boldsymbol{\xi}_{t,i})$, and the weight matrix, $W$; importantly, the updates of the former (winning unit posteriors) regulate the updates of the latter (weight matrix).

A snapshot of a complete TWTA layer is provided in Fig. 3 of Supplementary Section D.

**Inference**. As we depict in Fig. 1, during inference for a given task $t$, we retain the unit with maximum hidden winner indicator variable posterior, $\pi_{t,i,j}$, in each TWTA block $i$, and prune-out the weights pertaining to the remainder of the network. The feedforward pass is performed, by definition, by computing the task-wise discrete masks:

$$\tilde{\boldsymbol{\xi}}_{t,i} = \text{onehot}\left(\arg\max_j \pi_{t,i,j}\right) \in \mathbb{R}^J. \text{ Thus: winner}_{t,i} \triangleq \arg\max_j \pi_{t,i,j}. \tag{3}$$

Apparently, this way the proportion of retained weights for task $t$ is only equal to the $\frac{1}{J} * 100\%$ of the number of weights the network is initialized with.

### 2.3 A CONVOLUTIONAL VARIANT

Further, to accommodate architectures comprising convolutional operations, we consider a variant of the TWTA layer, inspired from Panousis et al. (2019). In the remainder of this work, this will be referred to as the Conv-TWTA layer, while the original TWTA layer will be referred to as the dense variant. The graphical illustration of Conv-TWTA is provided in Fig. 2.

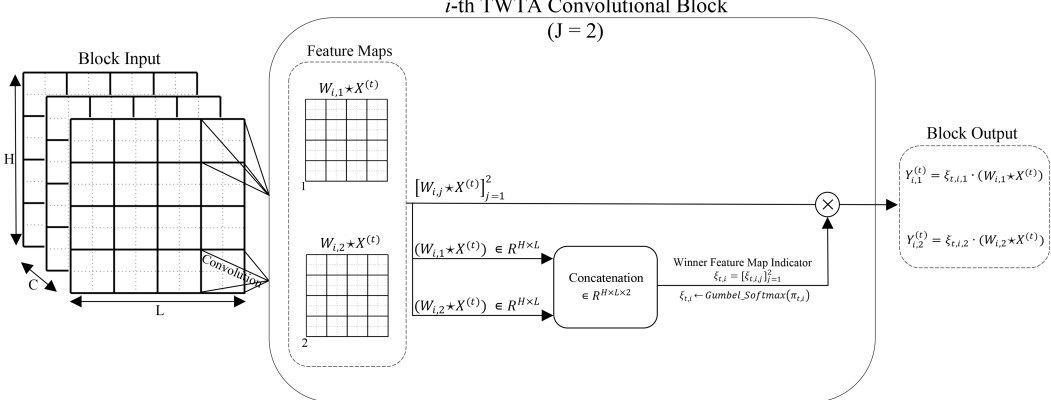

Figure 2: The convolutional TWTA variant (Section 2.3); for demonstration purposes, we choose $J = 2$ competing feature maps per kernel. Due to the TWTA mechanism, during forward passes through the network, only one competing feature map propagates its output to the next layer; the rest are zeroed-out.

Specifically, let us assume an input tensor $\boldsymbol{X}^{(t)} \in \mathbb{R}^{H \times L \times C}$ of a layer, where $H, L, C$ are the height, length and channels of the input. We define a set of kernels, each with weights $\boldsymbol{W}_i \in \mathbb{R}^{h \times l \times C \times J}$, where $h, l, C, J$ are the kernel height, length, channels and competing feature maps, and $i = 1, \ldots, I$.

Here, analogously to the grouping of linear units in a dense TWTA layer of Section 2.2, local competition is performed among feature maps in a kernel. Thus, each kernel is treated as a TWTA

block, feature maps in a kernel compete among them, and multiple kernels of competing feature maps constitute a Conv-TWTA layer.

This way, the output $\boldsymbol{Y}^{(t)} \in \mathbb{R}^{H \times L \times (I \cdot J)}$ of a layer under the proposed convolutional variant is obtained via concatenation along the last dimension of the subtensors $\boldsymbol{Y}_i^{(t)}$:

$$\boldsymbol{Y}_i^{(t)} = \boldsymbol{\xi}_{t,i} \cdot (\boldsymbol{W}_i \star \boldsymbol{X}^{(t)}) \in \mathbb{R}^{H \times L \times J} \tag{4}$$

where "$\star$" denotes the convolution operation.

Here, the winner indicator variables $\boldsymbol{\xi}_{t,i}$ are drawn again from the distribution of Eq. (2); they now govern competition among feature maps of a kernel.

**Inference**. At inference time, we employ a similar winner-based weight pruning strategy as for dense TWTA layers; for each task, the weights associated with one feature map (the winner) are retained while the rest are zeroed-out. Specifically, the winning feature map in a kernel $i$ for task $t$ is selected through $\arg\max$ over the hidden winner indicator variable posteriors, $\pi_{t,i,j} \; \forall j$, similar to Eq. (3). (see also Fig. 2).

## 2.4 TRAINING

For each task $t$, our approach consists in executing *a single* full training cycle. The performed training cycle targets both the network weights, $\boldsymbol{W}$, and the posterior hyperparameters $\boldsymbol{\pi}_{t,i}$ of the winner indicator hidden variables pertaining to the task, $p(\boldsymbol{\xi}_{t,i}) \; \forall i$.

The vectors $\boldsymbol{\pi}_{t,i}$ are initialized at random, while the network weights, $\boldsymbol{W}$, "continue" from the estimator obtained after training on task $t-1$. We denote as $\boldsymbol{W}^{(t)}$ the updated weights estimator obtained through the training cycle on the $t$-th task.

To perform training, we resort to maximization of the resulting evidence lower-bound (ELBO) of the network. Let us consider the $u$-th training iteration on task $t$, with data batch $D_u^{(t)} = (X_u^{(t)}, Y_u^{(t)})$. The training criterion is defined as:

$$\begin{aligned} L_u^{(t)} = &-\mathrm{CE}(Y_u^{(t)}, f(X_u^{(t)}; \boldsymbol{W}^{(t)}, \hat{\boldsymbol{\xi}}_t)) - KL[\, p(\boldsymbol{\xi}_t) \,\|\, q(\boldsymbol{\xi}_t)\,] \\ &- KL[\, f(X_u^{(t-1)}; \boldsymbol{W}^{(t)}, \tilde{\boldsymbol{\xi}}_{t-1}) \,\|\, f(X_u^{(t-1)}; \boldsymbol{W}^{(t-1)}, \tilde{\boldsymbol{\xi}}_{t-1})\,] - KD_{loss} \end{aligned} \tag{5}$$

Here, $\mathrm{CE}(Y_u^{(t)}, f(X_u^{(t)}; \boldsymbol{W}^{(t)}, \hat{\boldsymbol{\xi}}_t))$ is the categorical cross-entropy between the data labels $Y_u^{(t)}$ and the class probabilities $f(X_u^{(t)}; \boldsymbol{W}^{(t)}, \hat{\boldsymbol{\xi}}_t)$ generated from the penultimate Softmax layer of the network; $\hat{\boldsymbol{\xi}}_t = [\hat{\boldsymbol{\xi}}_{t,i}]_{i=1}^{I}$ is a vector concatenation of single Monte-Carlo (MC) samples drawn from the Categorical posteriors $p(\boldsymbol{\xi}_{t,i})$.

To ensure low-variance ELBO gradients with only one drawn MC sample from each posterior, we reparameterize these samples by resorting to the Gumbel-Softmax relaxation (Maddison et al., 2017). The Gumbel-Softmax relaxation yields sampled instances $\hat{\boldsymbol{\xi}}_{t,i}$ under the following expression:

$$\begin{aligned} \hat{\boldsymbol{\xi}}_{t,i} &= \mathrm{Softmax}(([\log \pi_{t,i,j} + g_{t,i,j}]_{j=1}^{J})/\tau) \in \mathbb{R}^J, \; \forall i \\ &\text{where}: g_{t,i,j} = -\log(-\log U_{t,i,j}), \; U_{t,i,j} \sim \mathrm{Uniform}(0,1) \end{aligned} \tag{6}$$

and $\tau \in (0, \infty)$ is a temperature factor that controls how closely the Categorical distribution $p(\boldsymbol{\xi}_{t,i})$ is approximated by the continuous relaxation.

Further, we consider that, a priori, $q(\boldsymbol{\xi}_{t,i})$ is a Categorical$(1/J)$; then, we obtain:

$$\begin{aligned} KL[\, p(\boldsymbol{\xi}_{t,i}) \,\|\, q(\boldsymbol{\xi}_{t,i})\,] &= \mathbb{E}_{p(\boldsymbol{\xi}_{t,i})}[\log p(\boldsymbol{\xi}_{t,i}) - \log q(\boldsymbol{\xi}_{t,i})] \approx \log p(\hat{\boldsymbol{\xi}}_{t,i}) - \log q(\hat{\boldsymbol{\xi}}_{t,i}) \Rightarrow \\ KL[\, p(\boldsymbol{\xi}_t) \,\|\, q(\boldsymbol{\xi}_t)] &= \sum_{i=1}^{I} \left( \log p(\hat{\boldsymbol{\xi}}_{t,i}) - \log q(\hat{\boldsymbol{\xi}}_{t,i}) \right) \end{aligned} \tag{7}$$

where we use Eq. (6), that is the Gumbel-Softmax relaxation, $\hat{\boldsymbol{\xi}}_{t,i}$, of the Categorical hidden winner indicator variables $\boldsymbol{\xi}_{t,i}$. This is similar to Panousis et al. (2019).

In addition, we have augmented our loss function with the KL divergence between the pairs of Softmax logits pertaining to $(t-1)$-th task data, obtained by using the winner masks $\tilde{\boldsymbol{\xi}}_{t-1,i}$ of the $(t-1)$-th task, defined in Eq. (3), and either: (i) the weight updates $\boldsymbol{W}^{(t-1)}$ at the end of the training cycle on the $(t-1)$-th task; or (ii) the unknown weight updates $\boldsymbol{W}^{(t)}$ sought on the $t$-th task. This prods the network towards retaining the logits of the previous task during network parameters updating. Thus:

$$KL[\,f(X_u^{(t-1)};\boldsymbol{W}^{(t)},\tilde{\boldsymbol{\xi}}_{t-1})\,||\,f(X_u^{(t-1)};\boldsymbol{W}^{(t-1)},\tilde{\boldsymbol{\xi}}_{t-1})\,] = \log f(X_u^{(t-1)};\boldsymbol{W}^{(t)},\tilde{\boldsymbol{\xi}}_{t-1}) - \log f(X_u^{(t-1)};\boldsymbol{W}^{(t-1)},\tilde{\boldsymbol{\xi}}_{t-1}) \tag{8}$$

where we use Eq. (3), that is the pruned network pertaining to task $(t-1)$, obtained through application of the binary masks $\tilde{\boldsymbol{\xi}}_{t-1,i}$.

Finally, in our experiments we employ an additional regularization term, similar to previous work, e.g. (Chen et al., 2021; Kang et al., 2023), to help our training process mitigate even more the effects of catastrophic forgetting. This regularization term is obtained as follows; consider task $t$:

1. In CIL, it is infeasible to have access to all data from the previous $t-1$ tasks. Thus, we randomly retain only few exemplars $X^{few}$ from the previous task, $t-1$, and store them in a limited size buffer.

2. These may be possibly augmented with more similar unlabeled data, $X'$, retrieved from some repository; selection is performed on the grounds of feature similarity with the data in the buffer, based on $l_2$ norm distance.

3. Then, we introduce an additional loss term that enforces similarity between the Softmax logits for $X^{KD} = (X^{few} \cup X')$ obtained by (i) using the old weight estimates, $\boldsymbol{W}^{(t-1)}$; and (ii) the sought updates, $\boldsymbol{W}^{(t)}$.

4. Following Hinton et al. (2015), we define the knowledge distillation loss in (5) as:

$$KD_{loss} = -H\left(f(X_u^{KD};\boldsymbol{W}^{(t)},\tilde{\boldsymbol{\xi}}_{t-1})\right) \cdot \log H\left(f(X_u^{KD};\boldsymbol{W}^{(t-1)},\tilde{\boldsymbol{\xi}}_{t-1})\right) \tag{9}$$

where we again use the pruned network pertaining to task $(t-1)$, while $H(f(\cdot)) \triangleq \text{Softmax}(f(\cdot)/\gamma)$ and $\gamma = 2$.

Note that, in our experiments, we also perform an ablation where the two aforementioned regularization terms of the training criterion (Eq. 5), defined in Eq's. (8) and (9), are omitted during training (see Supplementary Section C).

## 3 RELATED WORK

Recent works in (Chen et al., 2021; Kang et al., 2022; 2023) have pursued to build computationally efficient continual learners, that are robust to catastrophic forgetting, by drawing inspiration from LTH (Frankle & Carbin, 2019). These works compose sparse subnetworks that achieve comparable or/and even higher predictive performance than their initial counterparts. However, our work is substantially different from the existing state-of-the-art in various ways:
(i) Contrary to Chen et al. (2021), we do not employ iterative pruning, which repeats multiple full cycles of network training and pruning, until convergence. Instead, we perform a single training cycle, at the end of which we select a (task-specific) subnetwork to perform inference for the task.
(ii) Kang et al. (2022) and Kang et al. (2023) select a subnetwork that will be used for the task at hand on the grounds of an optimization criterion for binary masks imposed over the network weights. Once this subnetwork has been selected, Kang et al. (2022) proceeds to train the values of the retained weights while Kang et al. (2023) trains a randomly selected subset of the weights of the whole network, to account for the case of a suboptimal subnetwork selection. On the contrary, our method attempts to encourage different units in a competing block to specialize to different tasks. Training is performed concurrently for the winner unit indicator hidden variables, the posteriors of which regulate weight updates, as well as the network weights themselves. Thus, network pruning comes at the end of weight updating and not beforehand. We posit that this regulated updating scheme, which does not entail a priori hard pruning decisions, facilitates generalization without harming catastrophic forgetting.

## 4 EXPERIMENTS

We evaluate on CIFAR-100 (Krizhevsky et al., 2012), Tiny-ImageNet (Le & Yang, 2015), PMNIST (LeCun et al., 1998) and Omniglot Rotation (Lake et al., 2017). Also, we evaluate on the 5-Datasets (Saha et al., 2021) benchmark, in order to examine how our method performs in case that cross-task generalization concerns different datasets. We randomly divide the classes of each dataset into a fixed number of tasks with a limited number of classes per task. Specifically, in each training iteration, we construct $N$-way few-shot tasks by randomly picking $N$ classes and sampling few training samples for each class. In Supplementary Section A, we specify further experimental details for our datasets.

We adopt the original ResNet18 network (He et al., 2016) for Tiny-ImageNet, PMNIST and 5-Datasets; we use a 5-layer AlexNet similar to Saha et al. (2021) for the experiments on CIFAR-100, and LeNet (LeCun et al., 1998) for Omniglot Rotation. In the case of our approach, we modify those baselines by replacing each ReLU layer with a layer of (dense) TWTA blocks, and each convolutional layer with a layer of Conv-TWTA blocks. See more details in the Supplementary Section B.

For both the network weights, $\boldsymbol{W}$, and the log hyperparameters, $\log \boldsymbol{\pi}_{t,i}$, we employ Glorot Normal initialization (Glorot & Bengio, 2010). At the first training iteration of a new task, we initialize the Gumbel-Softmax relaxation temperature $\tau$ to 0.67; as the training proceeds, we linearly anneal its value to 0.01. We use SGD optimizer (Robbins, 2007) with a learning rate linearly annealed to 0, and initial value of 0.1. We run 100 training epochs per task, with batch size of 40.

### 4.1 EXPERIMENTAL RESULTS

Table 1: Comparisons on CIFAR-100, Tiny-ImageNet, PMNIST, Omniglot Rotation and 5-Datasets. We report the mean and standard deviation of the classification accuracy (%), obtained over three experiment repetitions with different seeds; * are results obtained from local replicates. We set $J = 8$; thus, the proportion of retained weights for each task, after training, is equal to the $(\frac{1}{J}*100 = 12.50)\%$ of the initial network. Also, we show the number of retained weights after training (in millions), for our method and the alternative approaches for reducing model size.

| Algorithm | CIFAR-100 | Tiny-ImageNet | PMNIST | Omniglot Rotation | 5-Datasets |
|---|---|---|---|---|---|
| GEM (Lopez-Paz & Ranzato, 2017) | 59.24 ±0.61* | 39.12 ±0.72* | - | - | - |
| iCaRL (Rebuffi et al., 2017) | 42.45 ±0.27* | 43.97 ±0.40* | 55.82 ±0.71* | 44.60 ±0.61* | 48.01 ±1.03* |
| ER (Chaudhry et al., 2019) | 59.12 ±0.82* | 37.65 ±0.88* | - | - | - |
| IL2M (Belouadah & Popescu, 2019) | 53.24 ±0.33* | 47.13 ±0.49* | 60.12 ±1.05* | 51.31 ±0.51* | 55.93 ±0.46* |
| La-MAML (Gupta et al., 2020) | 60.02 ±1.08* | 55.45 ±1.10* | 80.82 ±0.23* | 61.73 ±0.31* | 75.92 ±1.04* |
| FS-DGPM (Deng et al., 2021) | 63.81 ±0.22* | 59.74 ±0.64* | 80.92 ±0.63* | 62.83 ±0.93* | 76.10 ±0.13* |
| GPM (Saha et al., 2021) | 62.40 ±0.09* | 56.28 ±0.81* | 83.51 ±0.11* | 74.63 ±0.60* | 80.75 ±0.67* |
| SoftNet (80%, 4.5M params) | 48.52 ±0.08* | 54.02 ±1.13* | 64.02 ±0.38* | 55.82 ±0.29* | 57.60 ±1.12* |
| SoftNet (10%, 0.69M params) | 43.61 ±0.18* | 47.30 ±0.92* | 57.93 ±0.65* | 46.83 ±1.26* | 52.11 ±1.40* |
| LLT (100%, 11M params) | 61.46 ±1.18* | 58.45 ±0.73* | 80.38 ±0.08* | 70.19 ±0.24* | 74.61 ±0.81* |
| LLT (6.87%, 0.77M params) | 62.69 ±0.05* | 59.03 ±0.57* | 80.91 ±1.02* | 68.46 ±1.04* | 75.13 ±0.48* |
| WSN (50%, 4.2M params) | 64.41 ±0.60* | 57.83 ±0.94* | 84.69 ±0.18* | 73.84 ±0.35* | 82.13 ±0.04* |
| WSN (8%, 0.68M params) | 63.24 ±0.34* | 57.11 ±0.63* | 83.03 ±1.04* | 72.91 ±0.19* | 79.61 ±0.10* |
| **TWTA-CIL (12.50%, 0.39M params)** | **66.53** ±0.42 | **61.93** ±0.11 | **85.92** ±0.13 | **76.48** ±0.62 | **83.77** ±0.50 |

In Table 1, we show how TWTA-CIL performs in various benchmarks compared to popular alternative methods. We emphasize that the performance of SoftNet and WSN is provided for the configuration reported in the literature that yields the best accuracy, as well as for the reported configuration that corresponds to the proportion of retained weights closest to our method. Turning to LLT, we report how the method performs with no pruning and with pruning ratio closest to our method.

As we observe, our method outperforms the existing state-of-the-art in every considered benchmark. For instance, WSN performs worse than TWTA-CIL (12.50%), irrespectively of whether WSN retains a greater or a lower proportion of the initial network. Thus, our approach successfully discovers sparse subnetworks (*winning tickets*) that are powerful enough to retain previous knowledge, while generalizing well to new unseen tasks. Finally, it is interesting to examine how the winning ticket vectors differentiate across tasks. To this end, we compute the overlap among the $\tilde{\boldsymbol{\xi}}_t = [\tilde{\boldsymbol{\xi}}_{t,i}]_i$ vectors, defined in Eq. (3), for all consecutive pairs of tasks, $(t-1, t)$, and compute average percentages. We observe that average overlap percentages range from 6.38% to 10.67% across the considered datasets; this implies clear differentiation.

### 4.1.1 COMPUTATIONAL TIMES FOR TRAINING CIL METHODS

In Table 2, we report training wall-clock times for TWTA-CIL and the locally reproduced state-of-the-art alternatives. Our experiments have been executed on a single Tesla-K40c GPU. It is apparent that our method yields much improved training algorithm computational costs over all the alternatives.

Table 2: Average training wall-clock time (in secs), c.f. Table 1.

| Algorithm | CIFAR-100 | Tiny-ImageNet | PMNIST | Omniglot Rotation | 5-Datasets |
|---|---|---|---|---|---|
| GEM | 3747.15 | 6856.32 | - | - | - |
| iCaRL | 1350.68 | 2449.05 | 1278.38 | 150.73 | 3404.49 |
| ER | 3869.78 | 7182.04 | - | - | - |
| IL2M | 3896.67 | 7487.16 | 2767.11 | 323.59 | 7924.12 |
| La-MAML | 4073.61 | 7560.34 | 3726.18 | 443.91 | 10430.52 |
| FS-DGPM | 4398.16 | 7839.63 | 4012.36 | 471.03 | 11323.63 |
| GPM | 4104.05 | 7481.03 | 3949.56 | 470.70 | 9803.71 |
| SoftNet (80%) | 4791.13 | 7984.05 | 3602.84 | 419.28 | 9760.44 |
| SoftNet (10%) | 3091.16 | 5529.12 | 2314.67 | 283.06 | 6412.02 |
| LLT (100%) | 3648.82 | 6746.32 | 2719.61 | 316.91 | 7746.85 |
| LLT (6.87%) | 1850.36 | 3449.52 | 1268.50 | 155.53 | 3613.81 |
| WSN (50%) | 4952.67 | 8074.94 | 3679.73 | 443.06 | 10413.92 |
| WSN (8%) | 3262.04 | 5915.86 | 2415.57 | 296.14 | 6837.21 |
| **TWTA-CIL (12.50%)** | **1039.73** | **1914.63** | **859.27** | **127.41** | **2512.74** |

### 4.1.2 REDUCTION OF FORGETTING TENDENCIES

To examine deeper the obtained improvement in forgetting tendencies, we report the *backward-transfer and interference* (BTI) values of the considered methods in Table 3. BTI measures the average change in the accuracy of each task from when it was learnt to the end of the training, that is training on the last task; thus, it is immensely relevant to this empirical analysis. A smaller value of BTI implies lesser forgetting as the network gets trained on additional tasks. As Table 3 shows, our approach forgets less than the baselines on all benchmarks.

Table 3: BTI over the considered algorithms and datasets of Table 1; the lower the better.

| Algorithm | CIFAR-100 | Tiny-ImageNet | PMNIST | Omniglot Rotation | 5-Datasets |
|---|---|---|---|---|---|
| iCaRL | 13.41 | 6.45 | 8.51 | 18.41 | 23.56 |
| IL2M | 20.41 | 7.61 | 9.03 | 14.60 | 19.14 |
| La-MAML | 7.84 | 13.84 | 10.51 | 17.04 | 15.13 |
| FS-DGPM | 9.14 | 12.25 | 8.85 | **13.64** | 19.51 |
| GPM | 12.44 | 8.03 | 11.94 | 16.39 | 17.11 |
| SoftNet (80%) | 13.80 | 9.62 | 10.38 | 18.12 | 18.04 |
| SoftNet (10%) | 12.09 | 8.33 | 9.76 | 16.30 | 18.68 |
| LLT (100%) | 15.02 | 7.05 | 9.54 | 15.31 | 14.80 |
| LLT (6.87%) | 14.61 | 3.51 | 11.84 | 17.12 | 17.46 |
| WSN (50%) | 11.14 | 4.81 | 10.51 | 14.20 | 20.41 |
| WSN (8%) | 10.58 | 8.78 | 9.32 | 15.34 | 18.92 |
| **TWTA-CIL (12.50%)** | **6.14** | **2.50** | **8.04** | **13.64** | **13.51** |

## 4.2 ABLATIONS

### 4.2.1 EFFECT OF BLOCK SIZE $J$

Further, we re-evaluate TWTA-CIL with various block size values $J$ (and correspondingly varying number of layer blocks, $I$). In all cases, we ensure that the total number of feature maps, for a convolutional layer, or units, for a dense layer, which equals $I * J$, remains the same as in the original architecture of Section 4.1. This is important, as it does not change the total number of trainable parameters, but only the organization into blocks under the local winner-takes-all rationale. Different selections of $J$ result in different percentages of remaining network weights at inference time, as we can see in Table 4 (datasets Tiny-ImageNet and CIFAR-100). As we observe, the "TWTA-CIL (12.50%)" alternative, with $J = 8$, is the most accurate configuration of TWTA-CIL.

Table 4: Effect of block size $J$; Tiny-ImageNet and CIFAR-100 datasets. The higher the block size $J$ the lower the fraction of the trained network retained at inference time.

| | **Tiny-ImageNet** | | | **CIFAR-100** | | |
|---|---|---|---|---|---|---|
| **Algorithm** | Time | Accuracy | J | Time | Accuracy | J |
| TWTA-CIL (50%) | 2634.02 | 61.32 ± 0.28 | 2 | 1493.79 | 65.73 ± 0.20 | 2 |
| TWTA-CIL (25%) | 2293.81 | 61.04 ± 0.13 | 4 | 1301.20 | 65.45 ± 0.10 | 4 |
| **TWTA-CIL (12.50%)** | 1914.63 | **61.93 ± 0.11** | 8 | 1039.73 | **66.53 ± 0.42** | 8 |
| TWTA-CIL (6.25%) | 1556.09 | 61.45 ± 0.12 | 16 | 801.46 | 65.89 ± 0.12 | 16 |
| TWTA-CIL (3.125%) | 1410.64 | 60.86 ± 0.55 | 32 | 785.93 | 65.40 ± 0.48 | 32 |

### 4.2.2 EXPERIMENTAL RESULTS ON TASK-INCREMENTAL LEARNING

Finally, we repeat the experiments on all datasets of Table 1 in a task-incremental learning (TIL) setting. Such a setting requires the correct task identifier (task-id) for each test sample to be given at inference. This means that: (i) at training time, we update only the latent indicator vector $\xi$ that pertains to the known task at hand; (ii) at inference time, we use the inferred latent indicator vector $\xi$ of the task we are dealing with, which was obtained at a previous training cycle. In all cases, we use the last update of the network weights $W$.

In Table 5, we show how TWTA-CIL performs on CIFAR-100, Tiny-ImageNet, PMNIST, Omniglot Rotation, and 5-Datasets and compare our findings to current state-of-the-art algorithms; † denotes results adopted from Kang et al. (2022), while * are results obtained from local replicates. For the latter, the reported results are averages of the classification (%) accuracy and corresponding standard deviations obtained over three experiment repetitions with different random seeds. As we observe, our method outperforms the existing state-of-the-art in every considered benchmark.

Table 5: Performance in the benchmarks of Table 1 when task-id is known, i.e. a TIL scenario.

| Algorithm | CIFAR-100 | Tiny-ImageNet | PMNIST | Omniglot Rotation | 5-Datasets |
|---|---|---|---|---|---|
| EWC (Kirkpatrick et al., 2017) | 72.77 ± 0.45† | - | 92.01 ± 0.56† | 68.66 ± 1.92† | 88.64 ± 0.26† |
| GEM (Lopez-Paz & Ranzato, 2017) | 70.15 ± 0.34† | 50.57 ± 0.61† | - | - | - |
| iCaRL (Rebuffi et al., 2017) | 53.50 ± 0.81† | 54.77 ± 0.32† | 66.89 ± 0.82* | 53.07 ± 1.13* | 62.13 ± 0.71* |
| ER (Chaudhry et al., 2019) | 70.07 ± 0.35† | 48.32 ± 1.51† | - | - | - |
| La-MAML (Gupta et al., 2020) | 71.37 ± 0.67† | 66.90 ± 1.65† | 90.02 ± 0.11* | 70.62 ± 1.73* | 86.92 ± 1.31* |
| FS-DGPM (Deng et al., 2021) | 74.33 ± 0.31† | 70.41 ± 0.30† | 91.56 ± 0.19* | 73.09 ± 1.14* | 87.90 ± 0.33* |
| GPM (Saha et al., 2021) | 73.18 ± 0.52† | 67.39 ± 0.47† | 94.96 ± 0.07† | 85.24 ± 0.37† | 91.22 ± 0.20† |
| SoftNet (80%) | 60.13 ± 0.41* | 63.81 ± 1.16* | 75.30 ± 0.11* | 62.04 ± 0.84* | 70.38 ± 0.17* |
| SoftNet (10%) | 55.01 ± 1.40* | 57.15 ± 0.28* | 68.54 ± 1.12* | 55.92 ± 0.04* | 66.01 ± 1.54* |
| LLT (100%) | 72.65 ± 0.24* | 69.13 ± 0.51* | 92.80 ± 0.65* | 82.30 ± 0.28* | 86.42 ± 0.71* |
| LLT (6.87%) | 73.08 ± 0.17* | 70.15 ± 0.18* | 92.12 ± 0.33* | 81.63 ± 0.35* | 86.03 ± 0.51* |
| WSN (50%) | 76.38 ± 0.34† | 69.06 ± 0.82* | 96.24 ± 0.11† | 79.80 ± 2.16† | 93.41 ± 0.13* |
| WSN (8%) | 74.12 ± 0.37* | 71.40 ± 0.12* | 95.80 ± 0.32* | 83.72 ± 0.28* | 92.13 ± 0.07* |
| **TWTA-CIL (12.50%)** | **77.61 ± 0.23** | **72.28 ± 0.19** | **96.43 ± 0.03** | **86.53 ± 0.18** | **93.81 ± 0.24** |

## 5 CONCLUSION

In this work, we presented a novel CIL method called TWTA-CIL. Different from recent works, we devised stochastic TWTA activations that produce task-specific representations that are sparse and stochastic; the goal was to attain greater generalization ability across incrementally learned tasks. From our empirical analysis, the devised task-wise winning ticket formulation appears to successfully identify sparse subnetworks, preserving accuracy much better than the alternatives. Notably, TWTA-CIL prunes the network well, training requires less computational budget, and forgetting is significantly improved. Finally, our method retains its competitive advantage in TIL settings as well.

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

## A    MORE DETAILS ON THE USED DATASETS

**Datasets and Task Splittings**   5-Datasets is a mixture of 5 different vision datasets: CIFAR-10, MNIST (LeCun et al., 1998), SVHN (Netzer et al., 2011), FashionMNIST (Xiao et al., 2017) and notMNIST (Bui & Chang, 2016). Each dataset consists of 10 classes, and classification on each dataset is treated as a single task. PMNIST is a variant of MNIST, where each task is generated by shuffling the input image pixels by a fixed permutation. In the case of Omniglot Rotation, we preprocess the raw images of Omniglot dataset by generating rotated versions of $(90°, 180°, 270°)$ as in Kang et al. (2022). For 5-Datasets, similar to Kang et al. (2022), we pad 0 values to raw images of MNIST and FashionMNIST, convert them to RGB format to have a dimension of 3×32×32, and finally normalize the raw image data. All datasets used in Section 4 were randomly split into training and testings sets with ratio of 9:1. The number of stored images in the memory buffer - per class - is 5 for Tiny-ImageNet, and 10 for CIFAR-100, PMNIST, Omniglot Rotation and 5-Datasets.

We randomly divide the 100 classes of CIFAR-100 into 10 tasks with 10 classes per task; the 200 classes of Tiny-ImageNet into 40 tasks with 5 classes per task; the 200 classes of PMNIST into 20 tasks with 10 classes per task; and in the case of Omniglot Rotation, we divide the available 1200 classes into 100 tasks with 12 classes per task. The $N$-way few-shot settings for the constructed tasks in each training iteration are: 10-way 10-shot for CIFAR-100, PMNIST and 5-Datasets, 5-way 5-shot for Tiny-ImageNet, and 12-way 10-shot for Omniglot Rotation.

**Unlabeled Dataset**   The external unlabeled data are retrieved from 80 Million Tiny Image dataset (Torralba et al., 2008) for CIFAR-100, PMNIST, Omniglot Rotation and 5-Datasets, and from ImageNet dataset (Krizhevsky et al., 2012) for Tiny-ImageNet. We used a fixed buffer size of 128 for querying the same number of unlabeled images per class of learned tasks at each training iteration, based on the feature similarity that is defined by $l_2$ norm distance.

## B    MODIFIED NETWORK ARCHITECTURE DETAILS

### B.1    RESNET18

The original ResNet18 comprises an initial convolutional layer with 64 3x3 kernels, 4 blocks of 4 convolutional layers each, with 64 3x3 kernels on the layers of the first block, 128 3x3 kernels for the second, 256 3x3 kernels for the third and 512 3x3 kernels for the fourth. These layers are followed by a dense layer of 512 units, a pooling and a final Softmax layer. In our modified ResNet18 architecture, we consider kernel size = 3 and padding = 1; in Table 6, we show the number of used kernels / blocks and competing feature maps / units, $J$, in each modified layer.

Table 6: Modified ResNet18 architecture parameters.

| Layer Type | Kernels / blocks $(J = 2)$ | Kernels / blocks $(J = 4)$ | Kernels / blocks $(J = 8)$ | Kernels / blocks $(J = 16)$ | Kernels / blocks $(J = 32)$ |
|---|---|---|---|---|---|
| TWTA-Conv | 8 | 4 | 2 | 1 | 1 |
| 4x TWTA-Conv | 8 | 4 | 2 | 1 | 1 |
| 4x TWTA-Conv | 8 | 4 | 2 | 1 | 1 |
| 4x TWTA-Conv | 16 | 8 | 4 | 2 | 1 |
| 4x TWTA-Conv | 16 | 8 | 4 | 2 | 1 |
| TWTA-Dense | 16 | 8 | 4 | 2 | 1 |

### B.2    ALEXNET

The 5-layer AlexNet architecture comprises 3 convolutional layers of 64, 128, and 256 filters with 4x4, 3x3, and 2x2 kernel sizes, respectively. These layers are followed by two dense layers of 2048 units, with rectified linear units as activations, and 2x2 max-pooling after the convolutional layers. The final layer is a fully-connected layer with a Softmax output. In our modified AlexNet architecture, we replace each dense ReLU layer with a layer of (dense) TWTA blocks, and each convolutional layer with a layer of Conv-TWTA blocks; in Table 7, we show the number of used kernels / blocks and competing feature maps / units, $J$, in each modified layer.

Table 7: Modified AlexNet architecture parameters.

| Layer Type | Kernels / blocks $(J = 2)$ | Kernels / blocks $(J = 4)$ | Kernels / blocks $(J = 8)$ | Kernels / blocks $(J = 16)$ | Kernels / blocks $(J = 32)$ |
|---|---|---|---|---|---|
| TWTA-Conv | 8 | 4 | 2 | 1 | 1 |
| TWTA-Conv | 8 | 4 | 2 | 1 | 1 |
| TWTA-Conv | 16 | 8 | 4 | 2 | 1 |
| TWTA-Dense | 64 | 32 | 16 | 8 | 4 |
| TWTA-Dense | 64 | 32 | 16 | 8 | 4 |

### B.3 LENET

The LeNet architecture comprises 2 convolutional layers of 20, and 50 feature maps, followed by one feedforward fully connected layer of 500 units, and a final Softmax layer. In our modified LeNet architecture, we replace each of the 2 convolutional layers with one layer of Conv-TWTA blocks; the former retains 2 kernels / blocks of 8 competing feature maps, and the latter 6 kernels / blocks of 8 competing feature maps. The fully connected layer is replaced with a dense TWTA-layer, consisting of 50 blocks of 8 competing units.

## C  ABLATION: HOW DOES TWTA-CIL PERFORM WITHOUT THE USE OF THE ADDITIONAL REGULARIZATION TERMS IN THE TRAINING CRITERION OF EQ. (5)?

In Table 8 below, we provide full experimental CIL results for our method without the use of the two regularization terms; namely, these two terms are the knowledge distillation term on an external unlabelled dataset (Eq. (9)) and the KL divergence term concerning the previous state of the model (Eq. (8)). As we show, omitting these terms incurs an accuracy drop which does not exceed $1\%$ in all cases. Thus, the two terms only offer an extra regularization option, which is of minor contribution to the overall method accuracy. In Table 9, we show how computational savings improve even more in that case.

Table 8: Comparisons on CIFAR-100, Tiny-ImageNet, PMNIST, Omniglot Rotation and 5-Datasets. We report the classification accuracies ($\%$) for our approach; † denotes results for our method without the use of the two regularization terms, while * are the results reported in Table 1.

| Algorithm | CIFAR-100 | Tiny-ImageNet | PMNIST | Omniglot Rotation | 5-Datasets |
|---|---|---|---|---|---|
| TWTA-CIL (12.50%) | 65.83† | 61.12† | 85.38† | 75.90† | 83.04† |
| **TWTA-CIL (12.50%)** | **66.53*** | **61.93*** | **85.92*** | **76.48*** | **83.77*** |

Table 9: Average training wall-clock time (in secs), c.f. Table 8; † denotes results for our method without the use of the two regularization terms, while * are the results reported in Table 2.

| Algorithm | CIFAR-100 | Tiny-ImageNet | PMNIST | Omniglot Rotation | 5-Datasets |
|---|---|---|---|---|---|
| TWTA-CIL (12.50%) | 987.04† | 1819.94† | 814.05† | 119.03† | 2387.94† |
| **TWTA-CIL (12.50%)** | **1039.73*** | **1914.63*** | **859.27*** | **127.41*** | **2512.74*** |

# D ILLUSTRATIVE EXAMPLE OF A PROPOSED TWTA LAYER

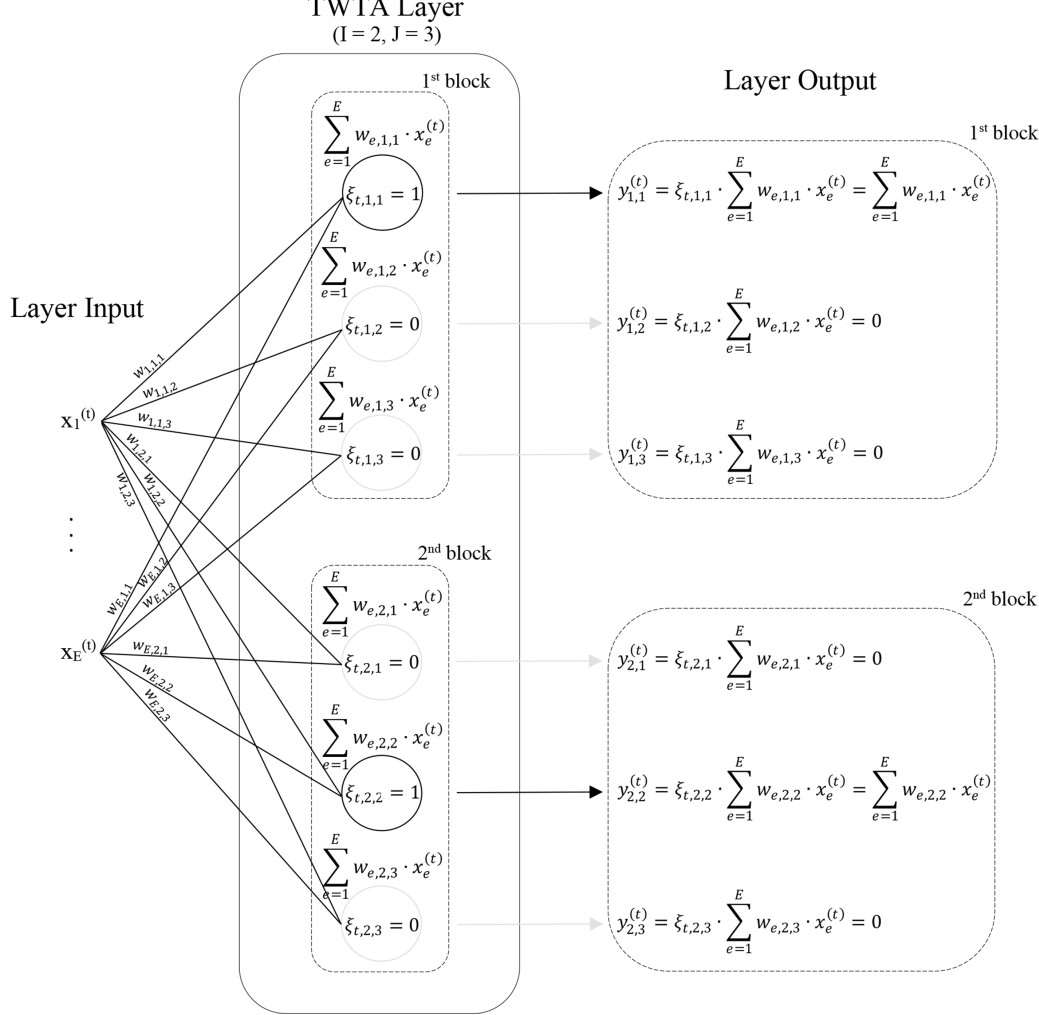

Figure 3: A high-level graphical illustration of a proposed TWTA layer; for demonstration purposes, we choose a TWTA layer composed of $I = 2$ blocks, with $J = 3$ competing units per block. Inputs $\boldsymbol{x}^{(t)} = \{x_1^{(t)}, \ldots, x_E^{(t)}\}$ are presented to each unit in the 2 blocks, when training on task $t$. Due to the TWTA mechanism, during forward passes through the network, only one competing unit propagates its output to the next layer; the rest are zeroed-out. For the demonstration purposes of the depicted example of the TWTA layer, we assume the first and second unit to be the winning units of the two blocks, respectively; only these units specialize in the $t$-th task.

