# OpenReview forum: "Continual Learning via Winning Subnetworks That Arise Through Stochastic Local Competition"
_ICLR.cc/2024/Conference — ICLR 2024 Conference Withdrawn Submission_

### Official Review · Reviewer_91vA · 2023-11-01

**Soundness:** 3 good
**Presentation:** 3 good
**Contribution:** 3 good
**Rating:** 6
**Confidence:** 3

**Summary:**

This paper describes a proposal for deep neural networks that use units that compete locally in a stochastic manner to represent new tasks. This approach creates sparse task-specific representations in each network layer, with different sparsity patterns for different tasks. During training, weight updates for each unit are regulated based on their winning probability. During inference, only the winning unit is retained, and the weights for non-winning units are set to zero for the given task. The authors claim to achieve state-of-the-art predictive accuracy in few-shot image classification experiments while requiring less computational resources compared to the current state-of-the-art techniques.

**Strengths:**

-	A competition strategy between units offers an exciting paradigm for continual learning. Although the use of subnetworks is not new, this competition strategy seems to work efficiently and reach good performance.
-	Overall, the computation time of the proposed approach is less than previous methods, achieving better performance.

**Weaknesses:**

-	My biggest concern is the scalability of this approach. Why are experiments with ImageNet100 not performed? It could be that the method is not suitable for this scale yet.
-	Could you add a column with the number parameter used by your method? If not, at least mention how it compares to other architectures.

**Questions:**

-	How much is the first task accuracy degrading compared to other methods? Is this competition strategy forgetting more or less in the first tasks? Does it have more recency bias?

---

> ### Author Response · Authors · 2023-11-17
> **Author Response to Reviewer 91vA**
>
> - "My biggest concern is the scalability of this approach. Why are experiments with ImageNet100 not performed? It could be that the method is not suitable for this scale yet." + "Could you add a column with the number parameter used by your method? If not, at least mention how it compares to other architectures." => In Table 1 of the revised paper, we have shown the number of retained parameters, after training, used by our method, and how it compares to the alternative approaches for reducing model size (namely SoftNet, LLT, and WSN). As we can see, the number of network parameters after training in our architecture is being reduced, compared to the aforementioned methods. Thus, our method is readily available to scale to ImageNet-100. Experiments on this dataset were not executed, because the main three competitors in our paper do not provide such experiments in their work.
>
> - "How much is the first task accuracy degrading compared to other methods? Is this competition strategy forgetting more or less in the first tasks? Does it have more recency bias?" => In Table 3 of our paper, we report the BTI values of our method and the other baselines. BTI is a standard metric, used to examine forgetting as the network gets trained on additional tasks. As Table 3 shows, our approach forgets less than the baselines on all benchmarks. Does the reviewer have something else in mind?

---

### Official Review · Reviewer_beRr · 2023-11-01

**Soundness:** 3 good
**Presentation:** 3 good
**Contribution:** 3 good
**Rating:** 6
**Confidence:** 3

**Summary:**

The authors propose a method for task incremental Continual Learning by leveraging sparse representing. The main idea is to group the neurons into blocks of ReLU units such that different units in a block learn different tasks and thus avoid interference while learning new tasks. During inference, only units corresponding to the task are used, and the rest are dropped, resulting in improved inference time. The evaluation against multiple baselines shows the efficacy of the method.

**Strengths:**

* The paper is well-written, and notations are easy to follow.

* The idea of grouping neurons into different blocks is novel and interesting. *Table 3** validates this and shows that the method can reduce forgetting between different tasks.

*The method is benchmarked against multiple baselines and shows substantial improvement, which confirms the hypothesis of the method.

* An ablation study to understand the effect of different block sizes is presented.

**Weaknesses:**

* > In our approach,  a group of J ReLU units is replaced by a group of J competing linear units, organized in one block; each layer contains I blocks of J

* It is unclear how the networks learn non-linearity if the ReLU units are removed.

* It would be interesting to see the comparison with dynamic sparse training-based and other sparsity-based methods is missing.

* > We keep few exemplars X few from the previous task, t − 1
It is not mentioned in the paper if the examples are selected randomly or based on some heuristic. It would be helpful to study the effect of exemplar selection on the method.

**Questions:**

* Why is *U* sampled from a Gaussian distribution instead of Gumbel distribution in **eqn 5**?

---

> ### Author Response · Authors · 2023-11-17
> **Author Response to Reviewer beRr**
>
> - "In our approach, a group of $J$ ReLU units is replaced by a group of $J$ competing linear units, organized in one block; each layer contains $I$ blocks of $J$. It is unclear how the networks learn non-linearity if the ReLU units are removed." => LWTA is another form of non-linearity. The non-linear nature is reduced by rectifying all units, but the winner.
>
> - "It would be interesting to see the comparison with dynamic sparse training-based and other sparsity-based methods is missing." => We have compared with other methods that induce sparsity [2,3,4]. Does the reviewer have some other method in mind ?
>
> - "We keep few exemplars $X^{few}$ from the previous task, $t-1$. It is not mentioned in the paper if the examples are selected randomly or based on some heuristic." => See response to reviewer RSpN (4th question of Section "Questions/Poor Presentation Quality").
>
> - "It would be helpful to study the effect of exemplar selection on the method." => For comparability issues, we used - similar to LLT [2] - a fixed memory buffer for obtaining few exemplars $X^{few}$ from the previous task, $t-1$. However, taking into account the paper length limitations, we commit to providing further ablations on exemplar selection impact in the final version of the Supplementary file; this will facilitate even further insights, beyond directly comparable experimental setups of different methods.
>
> - "Why is $U$ sampled from a Gaussian distribution instead of Gumbel distribution in Eq. (5)?" => $U$ is sampled from a continuous Uniform distribution, and not from a Gaussian distribution. That was originally defined by the Gumbel-Softmax equation, by [1].
>
> [1] Chris J. Maddison, Andriy Mnih, and Yee Whye Teh. The concrete distribution: A continuous relaxation of discrete random variables. In ICLR 2017.
>
> [2] T. Chen, Z. Zhang, S. Liu, S. Chang, and Z Wang. Long live the lottery: The existence of winning tickets in lifelong learning. In ICLR 2021.
>
> [3] Haeyong Kang, Rusty John Lloyd Mina, Sultan Rizky Hikmawan Madjid, Jaehong Yoon, Mark Hasegawa-Johnson, Sung Ju Hwang, and Chang D. Yoo. Forget-free continual learning with winning subnetworks. In ICML 2022.
>
> [4] Haeyong Kang, Jaehong Yoon, Sultan Rizky Hikmawan Madjid, Sung Ju Hwang, and Chang D. Yoo. On the soft-subnetwork for few-shot class incremental learning. In ICLR 2023.

---

> > ### Comment · Reviewer_beRr · 2023-11-22
> > **Re:**
> >
> > Thanks for addressing my concerns. I was referring to DST methods such as RigL, SET, etc.
> >
> > [1]. Rigging the Lottery: Making All Tickets Winners
> > [2]. Scalable training of artificial neural networks with adaptive sparse connectivity inspired by network science

---

> > > ### Author Response · Authors · 2023-11-23
> > > **Author Response to Reviewer beRr**
> > >
> > > - RigL [1] trains sparsely connected networks to reduce the amount of FLOPs, as opposed to our network organization that produces sparse representations and employs sparse training to prevent catastrophic overfitting. Thus, the two methods are not comparable, and such a comparison would be unfair and noncredible.
> > >
> > > - SET [2] starts from a random sparse topology (Erdös–Rényi random graph), and leads to a substantial reduction in connections, ending up with a trimmed network. However, in our work we do not trim the network, but obtain parts of it that are different among tasks. The whole network appears during our framework's processes, but different parts of it specialize in different tasks.

---

### Official Review · Reviewer_BkhS · 2023-11-08

**Soundness:** 3 good
**Presentation:** 2 fair
**Contribution:** 2 fair
**Rating:** 5
**Confidence:** 4

**Summary:**

The paper considers a deep network referred to as task winner-takes-all (TWTA), and it comprises unit blocks that compete locally to win the representation of each new task. The competition is performed in a stochastic manner (Gumbel Softmax), leading to regulating gradient-driven weight updates for a new task. During inference, the network retains only the winning unit and zeroes out all weights pertaining to the non-winning units for the task at hand. The proposed TWTA produces state-of-the-art predictive accuracy on class/task incremental tasks and imposes a considerably lower computational overhead than the current state-of-the-art.

**Strengths:**

-A novel mechanism referred to as task winner-takes-all (TWTA) that inherently learns to extract sparse task-specific data representations in a stochastic manner is proposed.
-The learned stochastic competition posteriors are used for regulating the weight training strength of TWTA.
-Winner-based weight pruning provided faster inference time.

**Weaknesses:**

-There is no analysis of sparse task-specific data representation and winning block units.
- Few-shot image classification tasks are mentioned; however, it is not explored in this work.
- The comparison of the number of parameters is not clear in architecture tables. Instead of talking about block/unit sizes, specific numbers would be easier to follow.

**Questions:**

(-) Does TWTA provide a forget-free solution in task-incremental learning (TIL)? How about the backward transfer?
(-) How sensitive is the performance to the selection? What is the performance when blocks are selected randomly?
(-) It would be nice to know the few-shot class incremental (FSCIL) performance of TWTA’s weight training strength regulation.

---

> ### Author Response · Authors · 2023-11-17
> **Author Response to Reviewer BkhS**
>
> - "There is no analysis of sparse task-specific data representation and winning block units." => As explained in the 'Inference' paragraph at the end of Section 2.2, sparsity is determined by the selection of number of competitors $J$. Also, we emphasize that we have shown a clear differentiation between the winning ticket vectors among tasks (see last paragraph of page 7 in the revised paper).
>
> - "Few-shot image classification tasks are mentioned; however, it is not explored in this work." => See response to reviewer RSpN (1st question of Section "Questions/Poor Presentation Quality").
>
> - "The comparison of the number of parameters is not clear in architecture tables. Instead of talking about block/unit sizes, specific numbers would be easier to follow." => In Table 1 of the revised paper, we provide the number of retained network parameters for our method and the alternative approaches for reducing model size.
>
> - "Does TWTA provide a forget-free solution in task-incremental learning (TIL)? How about the backward transfer?" => The knowledge distillation loss term in our training criterion provides the forget-free solution in the CIL settings, as we can see in Table 3. This same pattern is echoed in the case of TIL experiments. Due to space limitations, we commit to providing this ablation in the final version of the Supplementary file.
>
> - "How sensitive is the performance to the selection? What is the performance when blocks are selected randomly?" => We have noticed that the optimized masks $\\boldsymbol{\\xi}\_{t,i}$ differ from task to task. The essence of our method is the optimal selection of the hyperparameters $\\boldsymbol{\\pi}\_{t,i}$, so a random selection of units will not entail optimal performance; the performance drop in such a case varies in the range (20-40)%, depending on the dataset.
>
> - "It would be nice to know the few-shot class incremental (FSCIL) performance of TWTA’s weight training strength regulation." => This regulation is a consequence of the model formulation. Negating $\\hat{\\boldsymbol{\\xi}}\_{t,i}$ would offer a possibly interesting ablation, but we think it is not within the scope of this paper.

---

### Official Review · Reviewer_RSpN · 2023-11-09

**Soundness:** 2 fair
**Presentation:** 1 poor
**Contribution:** 3 good
**Rating:** 3
**Confidence:** 3

**Summary:**

The paper addresses the problem of catastrophic forgetting in class-incremental learning, focusing on finding specific subnetworks obtained by sparsifying the learned representation per task. Specifically, the authors remove the non-linearities in the neural network and reorganize the output of each hidden layer representation, such that it consists of a fixed number of blocks denoted as $I$ each characterized by $J$ output units.

The proposed approach involves learning a mask so that, at the end of each task training, only one out of $J$ output units per block is considered in the forward pass, reducing the overall number of parameters used during inference. Each task learns a separate mask, which are then at test time to evaluate continual learning performance. Specifically, for class incremental scenario only the last mask is used, while for task incremental the task-specific masks are used.

 To design  learnable masks, the authors use hidden winner indicator variables drawn from a Categorical posterior distribution. They  represent the proportion of output units to be masked in the current task for performing inference. The parameters of this distribution are learned by maximizing the Evidence Lower Bound, and sampling from this distribution is achieved through the Gumbel-Softmax trick, allowing backpropagation during training.

**Strengths:**

- Novel usage of Gumbel-Softmax trick to learn mask for identifying winning subnetworks in the class-incremental scenario.
- Their approach is supported by a good theoretical background and it seems well generalizing to large domain shift as proved by the experiments conducted on different datasets.
- The proposed approach can be plugged into various architecture based on linear and convolutional layers by simply swiching the standard linear and convolutional layers with the layers described in the paper.

**Weaknesses:**

From an high-level perspective, I have the following significant concerns:

* **Poor Presentation Quality**: the paper presentation is not optimal, and there are concerns about its clarity and organization.
* **Lack of Reproducibility**:  In the paper, many details necessary needed to reproduce the results are missing, such as a comprehensive list of the hyperparameters employed in the experiments related to the loss functions. While the code is included within the submission, clear instructions for its execution and for the dataset retrieval are needed.
* **Fixed-Size Classifier**: In a classical continual learning scenario, the prior knowledge of the number of classes to be encountered is not available. Common Continual Learning Frameworks (e.g., FACIL[1], Avalanche[2]) adapt the size of the last classifier as new classes are encountered, simulating a real scenario. It appears from the paper and the attached code that the output layer of the classifier is fixed from the start to the total number of classes that will be encountered. This represents a significant limitation of the proposed approach.
* **Scalability Limitations**: The paper lacks a dedicated section to discuss its scalability limitations, which is an important aspect for understanding the broader applicability of the proposed method.

[1]: Masana, M., et al. "Class-incremental learning: Survey and performance evaluation on image classification".

[2]: Carta, Antonio, et al. "Avalanche: A PyTorch Library for Deep Continual Learning"

**Questions:**

**Poor Presentation Quality**

* In the abstract, the authors claim that their approach achieves high predictive accuracy for few-shot image classification. However, the paper lacks experiments specifically focused on few-shot image classification. The only reference to few-shot class incremental learning coming from the citation of SoftNet (Kang et al. 2023), which actually tackles the problem of few-shot classification in incremental learning scenarios. Am I missing something? Have the author adapted the approach of Kang et al. to the standard class incremental learning to provide the results presented in Table 1?
* The Model Formulation section (Sec. 2.2) is challenging to follow. There is a need for a more detailed description of how the competing linear units are organized. To enhance clarity, I suggest to provide a high-level figure illustrating how the output units are organized and masked. This figure can be used as fundamental block for undestanding Figure 1 and Figure 2 and can help to follow the notation used in the subsequent sections. Moreover, increasing the font size of the mentioned figures will improve readability.
* To enhance the paper organization, it is advisable to improve image captions by directly referencing the corresponding sections of the paper. Additionally, positioning Figures and Tables at the top of the pages, rather than interrupting the text, is recommended to improve the readability.
* It would be beneficial to enhance the logical flow of the training section (Section 2.4). The current presentation gives the impression that each text block is somewhat disjointed from the preceding one. While the authors attempt to breakdown the training loss (Eq 7) into various components, the overall discussion is hard to follow. Since ELBO Maximization is not a standard procedure in the context of CIL scenario, I suggest introducing the ELBO formulation first, followed by how the training loss is linked to it and then presenting Equation 5. for Gumbel-Softmax relaxation. A reformulation of the section could help clearly delineate which terms in the loss are specifically tailored for learning the new task, which are added ad-hoc for continual learning regularization, and which originate from ELBO Maximization. Lastly, in Equation 6, the repetition of "J" within the parentheses might be unnecessary and the step in Equation (8) is not obvious, thus additional details may be provided.

  *Exemplars*: The concept of "Exemplars" is introduced twice in the text, before and after the training loss definition. Improving readability could involve describing the exemplar loss and how exemplars are managed in a reserved paragraph.  Moreover, considering the current organization, clarity is compromised in explaining how exemplars are managed (point 3.), as the description of the KD loss is only provided at the end of the section.

  *Supplementary References*: The paper lacks references to all the supplementary material sections.

**Training Loss Questions**:  I have some concerns regarding the training loss. Why does the third term depend on the dataset at task $t-1$ (i.e., $X_u^{(t-1)}$)? Additionally, it would be beneficial to gain a clearer understanding of the authors' choice to implement knowledge distillation, as described by Hinton et al., for exemplars, and to use the KL distance between network probabilities of the current task model and the previously trained one when dealing with the current task (the third component in the training loss). I wonder if there is a specific reason for not applying knowledge distillation when using data from the current task?

**Lack of Reproducibility**: The authors provide code without instructions on how to run it, and the current Readme.md comes from another work (Chen et al. [3]). Additionally, the existing code does not facilitate the execution of experiments outlined in the main paper:

* the standard dataset (Cifar10 ad Cifar100) are not automatically downloaded and should be downloaded manually from the official site. Even if this download is performed, the code for pre-processing the datasets is absent. Currently, the code requires that Cifar10 and Cifar100 must be organized in dictionaries already splitted in Train and Validation. The provided link in the Readme.md(from [3]) for pre-processed datasets is not working.
* The code doesn't support experiments on other datasets (Tiny-Imagenet, PMnist, Omniglot, 5-Datasets).
* Execution of experiments presented in the paper tables is not feasible. For example, the number of blocks $J$ is hard-coded and set to $2$ (while in the paper $J=8$  is indicated as the best-performing choice). Additionally, certain hyperparameters for training losses are hardcoded without mention in the main paper. For instance, a factor $\lambda=1/1000$ is placed in front of the KL distance between p and q (Equation 7, second component). See also next question.
* Hyperparameter Selection: What are the training loss hyperparameters and how they are selected? How  $\tau$ (Equation 5), the hyperparameter controlling the sparsity of the masks fixed to 0.67, is selected? A discussion in the main paper is required.
* Batch Size Dependency For Task-wise discrete mask. Does the size of parameters $\pi_{t, i, j}$ and the corresponding $\tilde{\xi}_{t,i}$ (Equation 6) depend on the batch size used during the training?

[3] Long Live the Lottery: The Existence of Winning Tickets in Lifelong Learning, Tianlong Chen\*, Zhenyu Zhang\*, Sijia Liu, Shiyu Chang, Zhangyang Wang

**Fixed-size Classifier**:  In addition to the concern previously raised about prior knowledge of the number of classes, I have two questions regarding the structure of the last classifier placed on the top of the network. In section 2.4, the authors mention that the class probabilities are generated by "the penultimate Softmax layer of the network." What does this mean? To the best of my knowledge, in the standard supervised classification paradigm for class incremental, a single Softmax function is typically applied after the last and only fully connected layer of the network. Moreover, inspecting the attached code, I see that in the file **small_model.py**, two final linear layers are defined (named *linear* and *linear_main*). Is this connected to the previous sentence? I'm curious to understand the rationale behind having two linear classification layers at the end of the network.


**Summary of the Review**: The paper presents a lot of interesting ideas with a good theoretical background in support. However, I have significant concerns on the reproducibility of the experiments and on the clarity of the presented method. I believe that the current version of the paper does not satisfy high quality standard for paper presention of ICLR. I am inclined to assign a reject score; however, I am open to reconsidering my decision during the rebuttal if the authors adequately address the issues raised for both the paper and the experimental code.

---

> ### Author Response · Authors · 2023-11-17
> **Author Response to Reviewer RSpN (part1)**
>
> Weaknesses:
>
> 1) "Lack of Reproducibility" => The requested details have been clearly encoded in the provided source code. We will expand on these in the final version of the Supplementary to facilitate the reader. Regarding the dataset retrieval, we have based our code on the source code of LLT [1]. For copyright issues, we cannot provide the datasets (and their splittings in train, validation, and test sets) and you have to obtain them directly from the authors of [1].
>
> 2) "Fixed-Size Classifier" => In each iteration of our CIL algorithm, the network is presented with training examples that pertain to a specific number of classes. Therefore, the rest of the penultimate Softmax layer is effectively not trained, but ignored. This is equivalent to the source code of FACIL and Avalanche, which add more output neurons as more tasks come into play. Thus, this difference is only a matter of code structure rather than algorithmic formulation. There is no limitation when it comes to the algorithm we propose, but only to our source code, which is indifferent, as we are not releasing industry-grade software but a rapid prototype for the purposes of our comparative evaluations.
>
> 3) "Scalability Limitations" => Not only is there no scalability drawback, but as we show in Table 2 of the main paper our approach imposes less computational burden compared to the alternatives. Regarding the total number of classes, we emphasise there is no actual need to know them a priori (see comment above).
>
> Questions:
>
> 1) "Poor Presentation Quality"
>   * We thank the reviewer for pointing out this omission. As one can see in the source code, we have used only few-shot versions of the mentioned datasets. Specifically, the $N$-way few-shot settings for the constructed tasks in each training iteration are: $10$-way $10$-shot for CIFAR-100, PMNIST and 5-Datasets, $5$-way $5$-shot for Tiny-ImageNet, and $12$-way $10$-shot for Omniglot Rotation. We are uploading a revised paper version, where we clarify this very important detail in the introduction of Section 4.
>
>   * Given that the other reviewers do not have the same worries, but to help the reviewer RSpN to get a better high-level understanding, we are providing a high-level figure illustrating how the output units are organized and masked (see the revised Supplementary material). Also, we increased the font size of Figures 1 and 2 to improve readability.
>
>   * In the uploaded version of the revised paper, we improved Fig.'s 1 and 2 captions by directly referencing the corresponding sections of the paper.
>
>   * ELBO maximization is standard in bleeding edge of deep learning, and especially class-incremental learning (e.g. [3]). In the revised paper, we restructured Section 2.4 to improve its readability and logical flow. Moreover, in the old Eq. (8), we draw only one MC sample $\\hat{\\boldsymbol{\\xi}}\_{t,i}$ from the Categorical posterior $p(\\boldsymbol{\\xi}\_{t,i})$, in order to compute the KL divergence between distributions $p(\\boldsymbol{\\xi}\_{t,i})$ and $q(\\boldsymbol{\\xi}\_{t,i})$. In the final step, we simply sum up all the $I$ KL divergence values.
>
>   * In the revised version of our paper, we included the references to all the supplementary material sections.
>
> 2) "Training Loss Questions" => Both the third and the fourth term help the model alleviate catastrophic forgetting. Since we cannot retain all past data, we use a dedicated term using all the data from the immediately previous task, as well as a final term using few randomly selected data from all previous tasks.
>
> 3) "Lack of Reproducibility"
>   * The code for the datasets pre-processing was already provided in the initial Supplementary material. However, we re-organize it in the revised version for all the datasets, to be even more clear. As for the datasets retrieval, see our comment in the "Lack of Reproducibility" Section of the reported Weaknesses above.
>
>   * We re-set the number of blocks $J$ as a configurable parameter throughout the code. Regarding the technicalities about the factor $\\lambda=\\frac{1}{1000}$, it is a standard practice to scale the KL divergence between two terms; otherwise, it will dominate the objective function.
>
>   * The selection of value $0.67$ for the Gumbel-Softmax relaxation temperature $\\tau$, is quite standard on the literature (e.g. [2]).
>
>   * The parameters $\\boldsymbol{\\pi}\_{t,i,j}$ and the corresponding $\\tilde{\\boldsymbol{\\xi}}\_{t,i}$ are task-wise, and not datapoint-wise. Therefore, there is no dependence on the batch size.

---

> ### Author Response · Authors · 2023-11-17
> **Author Response to Reviewer RSpN (part2)**
>
> 4) "Fixed-size Classifier" => The reviewer has probably misunderstood something about the usage of the two linear layers (named 'linear' and 'linear_main') in the file small_model.py. They are both conditional statements of an if-else case, so only one layer is used at each run. Thus, there is only 1 fully connected layer at the end of our network architecture. Also, by mentioning that the class probabilities are generated by "the penultimate Softmax layer of the network", we mean that after the last and only fully connected layer of our network, we apply a single Softmax layer in order to obtain the class probabilities.
>
> [1] T. Chen, Z. Zhang, S. Liu, S. Chang, and Z Wang. Long live the lottery: The existence of winning tickets in lifelong learning. In ICLR 2021.
>
> [2] Diederik P. Kingma, Max Welling. Auto-Encoding Variational Bayes. In ICLR 2014.
>
> [3] Gido M. van de Ven, Zhe Li, and Andreas S. Tolias. Class-Incremental Learning with Generative Classifiers. In CVPR 2021.

---

> > ### Comment · Reviewer_RSpN · 2023-11-22
> >
> > Thank you for your response. After carefully considering the author's rebuttal to my concerns and those raised by other reviewers, I am summarizing my decision.
> >
> > My primary concern revolves around the omission in the original submission regarding the task that the authors aim to solve, namely, few-shot class incremental learning. I believe that this initial omission is significant as it substantially alters the focus of the paper, making the review process more challenging. Furthermore, I still maintain that the presentation and reproducibility issues I raised in my review have not been adequately addressed. Just as an example, the question regarding the selection of hyperparameters remains unresolved. Lastly, I am skeptical about the feasibility of storing all the previous task data in class incremental learning, as indicated in the third component of the loss function.
> >
> > I acknowledge that the underlying idea is interesting and has the potential to become publishable in the future, but I still maintain my opinion that the paper should be rejected for this conference.

---

> > > ### Author Response · Authors · 2023-11-22
> > > **Author Response to Reviewer RSpN**
> > >
> > > - In the revised version of our paper (see first paragraph of Section 4), we have clearly clarified that we employed $N$-way few-shot settings for the constructed tasks in each training iteration. Also, in Supplementary Section A, we mention the specific settings used for each dataset, which are: 10-way 10-shot for CIFAR-100, PMNIST and 5-Datasets, 5-way 5-shot for Tiny-ImageNet, and 12-way 10-shot for Omniglot Rotation.
> > > - Regarding the hyperparameters omission, we would like to ask the reviewer to explain us which specific hyperparameter is not clearly defined in the revised version of our paper, after our rebuttal.
> > > - Lastly, the reviewer might have misunderstood the essence of our tactic regarding the data of the previous task. As we have clearly explained in Section 2.4 (see Eq. (8) that describes the third component of the loss function), we keep only the data $X\_u^{(t-1)}$ pertaining to the previous $(t-1)$-th task. Regarding the fourth component of the loss function (see Eq. (9)), we store only few exemplars $X^{few}$ from the previous task, $t-1$, in a limited size buffer. It is clear then that we do not keep any data from the previous $t-2$ tasks, as the reviewer falsely claims.

---

### Author Response · Authors · 2023-11-17
**Author Rebuttal by Authors**

We believe we have addressed all the issues raised by the reviewers. We would like to respectfully ask the reviewers to reconsider their scores. We are also happy to address any further questions.